# MLAE: Masked LoRA Experts for Visual Parameter-Efficient Fine-Tuning

## Abstract

In response to the challenges posed by the extensive parameter updates required for full fine-tuning of large-scale pre-trained models, parameter-efficient fine-tuning (PEFT) methods, exemplified by Low-Rank Adaptation (LoRA), have emerged. LoRA simplifies the fine-tuning process but may still struggle with a certain level of redundancy in low-rank matrices and limited effectiveness from merely increasing their rank. To address these issues, a natural idea is to enhance the independence and diversity of the learning process for the low-rank matrices. Therefore, we propose **M**asked **L**oRA **E**xperts (MLAE), an innovative approach that applies the concept of masking to visual PEFT. Our method incorporates a cellular decomposition strategy that transforms a low-rank matrix into independent rank-1 submatrices, or "experts", thus enhancing independence. Additionally, we introduce a binary mask matrix that selectively activates these experts during training to promote more diverse and anisotropic learning, based on expert-level dropout strategies. Our investigations reveal that this selective activation not only enhances performance but also fosters a more diverse acquisition of knowledge with a marked decrease in parameter similarity among MLAE, significantly boosting the quality of the model. Remarkably, MLAE achieves new state-of-the-art (SOTA) performance with an average accuracy score of 78.8% on the VTAB-1k benchmark and 90.9% on the FGVC benchmark, surpassing the previous SOTA result by an average of 0.8% on both benchmarks with approximately half parameters.

## 1 Introduction

Large-scale pre-trained vision models (Dosovitskiy et al., 2020; He et al., 2022; Liu et al., 2021; Zhai et al., 2022) have manifested outstanding performance across a variety of downstream vision tasks (Deng et al., 2009; Goyal et al., 2017; Kuehne et al., 2011; Khosla et al., 2011; Zhou et al., 2019). However, full parameter fine-tuning poses significant challenges to computational resources and data storage. To address this issue, considerable efforts have been invested in developing parameter-efficient fine-tuning (PEFT) methods (Chen et al., 2022; Jia et al., 2022; Jie & Deng, 2022; Zhang et al., 2022; Hu et al., 2021), which only fine-tunes a minimal number of parameters. Among these, Low-Rank Adaptation (LoRA) (Hu et al., 2021) stands out due to its simplicity and efficiency, which achieves high performance leveraging low-rank matrices. However, recent studies (Gao et al., 2024; Chen et al., 2023; Zoph et al., 2022) have indicated that even within low-rank matrices, a certain level of redundancy persists, and merely increasing the rank of the update matrix does not necessarily enhance the quality of the model. Consequently, how to simultaneously reduce redundancy in low-rank matrices and enhance their quality represents a fundamental challenge.

Recently, works exemplified by AdaLoRA (Zhang et al., 2023b) and IncreLoRA (Zhang et al., 2023a) have been proposed to dynamically adjust the matrix rank by adding or removing parameters in different layers of the model based on importance scores, addressing the issue of parameter redundancy in shallow networks. Inspired by these, we conceive the idea of temporarily removing certain parameters during the training phase, instead of permanently eliminating them without the chance of revitalization. This approach not only eliminates the need for calculating parameter scores, thereby reducing computational overhead, but also allows all parameters to participate during inference, enhancing the model's expressiveness and robustness. To this end, a natural idea is employing masking to the parameters. However, it is impractical to apply masking directly to parameters of a low-rank matrix, as it may result in training instability and inefficiency, while

individual parameters within such matrices do not carry any semantic information advocating diverse learning. We have observed that recent efforts (Zhu et al., 2023; Wu et al., 2023) have applied the LoRA framework within the mixture of experts (MoE) (Shazeer et al., 2017) architecture, where each LoRA weight is treated as an expert, and a gating mechanism selects experts to achieve sparsity. This motivates us to explore applying masking to LoRA experts, rather than directly to the parameters.

In this work, we propose a novel method - MLAE (Masked LoRA Experts), which applies the concept of masking to the field of visual parameter-efficient fine-tuning (PEFT) by enhancing the independence and diversity of learning to address the issue of parameter redundancy. Specifically, MLAE addresses redundancy and enhances the quality of LoRA from two perspectives: First, it employs a cellular decomposition approach, where the low-rank matrix with rank $r$ is further decomposed into $r$ rank-1 submatrices, which are treated as independent experts; Second, building on cellular decomposition, it further introduces a mask matrix with adaptive coefficients, applied to the decomposed expert matrix to implement MLAE. Through in-depth exploration of fixed, stochastic, and mixed masking with different configurations, we have discovered that the use of expert-level dropout to generate stochastic masks yields the best performance. Further analysis has revealed that the parameter similarity among these experts significantly decreases and they acquire a more diverse range of knowledge.

In summary, our contributions are mainly as follows:

- We have pioneered the introduction of masking strategies into the domain of visual PEFT, exploring various masking techniques including fixed, stochastic, and mixed masks. This innovation paves the way for future work in the field.

- We tackle an interesting and relevant issue in the field of fine-tuning vision pre-trained models, specifically the problem of parameter diversity and redundancy in LoRA, which is a significant contribution to the domain of parameter-efficient fine-tuning.

- We introduce MLAE, a method that is both simple and flexible. MLAE has attained state-of-the-art (SOTA) results on the VTAB-1k and FGVC benchmarks, achieving an average accuracy improvement of 0.8% over previous SOTA result on both benchmarks. This demonstrates its strong capabilities in the field of visual PEFT.

## 2 RELATED WORKS

**Parameter-Efficient Fine-Tuning.** As the number of model parameters has continued to grow, the significant computational and storage costs with the traditional full fine tuning have made Parameter-Efficient Fine-Tuning (PEFT) increasingly attractive. The PEFT methods for large-scale pre-trained models first emerged in the NLP field (Houlsby et al., 2019), where they achieve comparable performance to full fine tuning while only requiring the fine-tuning of a few lightweight modules. Inspired by this success in NLP, researchers have begun applying PEFT to Pretrained Vision Models (PVMs). Existing visual PEFT methods can be categorized into addition-based, unified-based, and partial-based methods. Addition-based methods (Jie & Deng, 2022; Liu et al., 2022; Dong et al., 2022; Jia et al., 2022; Chen et al., 2022; Li & Liang, 2021), exemplified by VPT (Jia et al., 2022), AdaptFormer (Chen et al., 2022), Fact Jie & Deng (2023) and Prefix-tuning (Li & Liang, 2021), incorporate additional trainable modules or parameters into the original PVMs to learn task-specific information. Unified-based methods (Yu et al., 2022; Zhang et al., 2022; Chavan et al., 2023; Jiang et al., 2023; Gao et al., 2023), represented by NOAH (Zhang et al., 2022), SPT (He et al., 2023) and GLoRA (Chavan et al., 2023), integrate various fine-tuning approaches into a single architecture. Partial-based methods (Fu et al., 2022; Hu et al., 2021; Houlsby et al., 2019; Lian et al., 2022; Luo et al., 2023), such as LoRA (Hu et al., 2021), Adapter (Houlsby et al., 2019), SSF (Lian et al., 2022), and RepAdapter (Luo et al., 2023), update only a small portion of the inherent parameters while keeping most of the model's parameters unchanged during the adaptation process. In this paper, we explore more efficient and high-quality strategies for visual PEFT based on partial-based methods.

**Masking Strategies.** In recent years, masking strategies have been extensively employed in deep learning, primarily focusing on self-supervised pre-training through mask modeling. Representative works (Bao et al., 2021; Zhou et al., 2021; Baevski et al., 2022; Geng et al., 2022; Arici et al., 2021; Kwon et al., 2022) include language model pre-training spearheaded by BERT (Devlin et al., 2018) and visual model pre-training exemplified by MAE (He et al., 2022) and SimMIM (Xie et al., 2022),

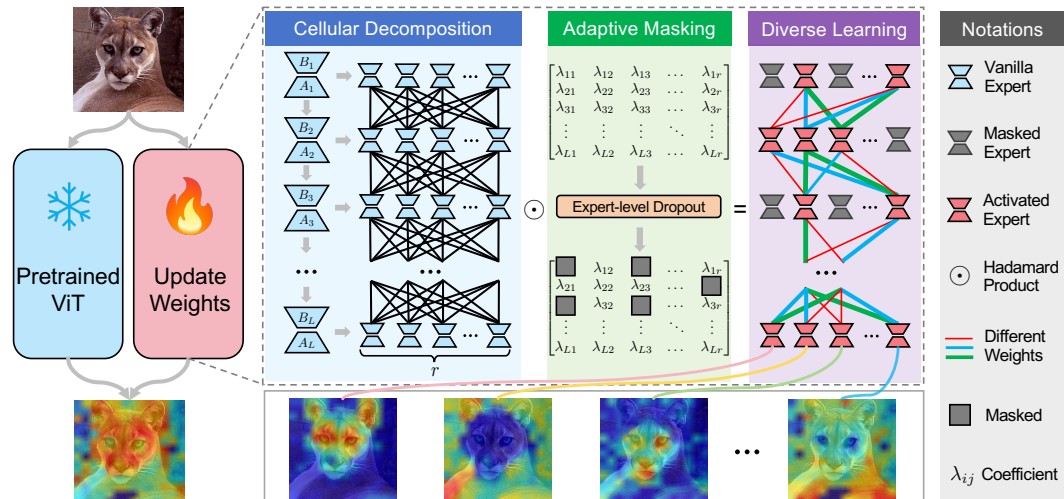

Figure 1: Our proposed Masked LoRA Experts (MLAE) framework.

both of which have achieved significant success. Typically, these works apply the masking strategy to the input sequences, retaining a portion of the input and training models to predict the masked content. Meanwhile, a smaller body of work (Książek & Spurek, 2023; Mallya & Lazebnik, 2018; Golkar et al., 2019; Serra et al., 2018; Chen et al., 2020; Kang et al., 2022) has applied masking strategies to the model architecture, achieving preliminary results in continual learning and multi-task learning. Diverging from these approaches, we innovatively apply the masking strategy to the lightweight modules of the model based on the PEFT methods, conducting a comprehensive and in-depth analysis that has yielded outstanding results.

## 3 METHODS

Unlike existing approaches that adjust the incremental matrix rank based on importance scores or select the optimal experts through gating mechanisms, our methodology aims to alleviate parameter redundancy by enforcing diversity in the learning process. The key idea of our methodology is to enhance the capabilities of the model within a given budget by promoting varied learning rather than merely eliminating underutilized parameters. Therefore, we focus on achieving independence and diversity in parameter learning through two strategic approaches: by designing innovative model architectures and introducing masking strategy for training. The complete process is shown in Fig. 1.

### 3.1 PRELIMINARY

**Low-Rank Adaptation.** LoRA (Hu et al., 2021) has been demonstrated to be a popular parameter-efficient fine-tuning approach to adapt pre-trained models to specific tasks. It leverages low-rank matrix decomposition of pre-trained weight matrices to significantly reduce the number of training parameters. Formally, for a pre-trained weight matrix $W_0 \in \mathbb{R}^{d_{\text{in}} \times d_{\text{out}}}$, LoRA creates two low-rank trainable matrices $A$ and $B$ with rank $r$, where $B \in \mathbb{R}^{d_{\text{in}} \times r}$, $A \in \mathbb{R}^{r \times d_{\text{out}}}$, and $r \ll \min\{d_{\text{in}}, d_{\text{out}}\}$.

$$h = W_0 x + \Delta W x = W_0 x + BA x \tag{1}$$

During training, LoRA updates only the two matrices $A$ and $B$, while $W_0$ is frozen. The matrix $A$ is initialized with a random Gaussian distribution and matrix $B$ is initialized to zero.

### 3.2 MASKED LoRA EXPERTS

Vanilla LoRA significantly reduces the fine-tuning parameters of over-parametrized models by updating only low-rank matrices, nevertheless, without carefully imposed independence and diversity constraints, a certain degree of redundancy may still remain within these low-rank matrices.

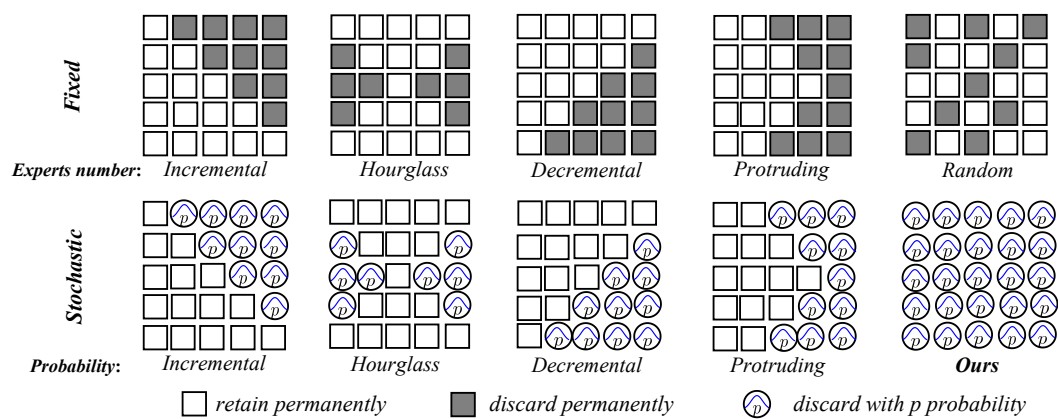

Figure 2: Different masking strategies.

**Cellular Decomposition.** To obtain independent experts, we propose to decompose the parameters of low-rank matrices into the disjoint union of cells, each represented by a rank-1 submatrix. We regard each rank-1 submatrix as a self-contained expert, thus the update matrix in the $l$-th layer is decomposed into an ensemble of $r$ experts. More specifically, the low-rank matrices $B$ and $A$ with rank $r$ in Equation (1) can be written as their respective column vectors $[b_1 \, b_2 \, \ldots \, b_r]^T$, and row vectors $[a_1 \, a_2 \, \ldots \, a_r]$. Correspondingly, the $BA$ matrices in the $l$-th layer can be expressed as: $\Delta W_l = \sum_{i=1}^r w_{li} = \sum_{i=1}^r b_{li} a_{li}$.

For a pre-trained model with a total of $L$ layers, each with $r$ experts, the model features $L \times r$ self-contained experts in total, denoted as expert matrix $\mathcal{E}$:

$$
\begin{bmatrix} \Delta W_1 \\ \Delta W_2 \\ \vdots \\ \Delta W_L \end{bmatrix} = \begin{bmatrix} \sum_{i=1}^r w_{1i} \\ \sum_{i=1}^r w_{2i} \\ \vdots \\ \sum_{i=1}^r w_{Li} \end{bmatrix} = \begin{bmatrix} \sum_{i=1}^r b_{1i} a_{1i} \\ \sum_{i=1}^r b_{2i} a_{2i} \\ \vdots \\ \sum_{i=1}^r b_{Li} a_{Li} \end{bmatrix} \xrightarrow{into\ matrix} \mathcal{E} = \begin{bmatrix} b_{11} a_{11} & b_{12} a_{12} & \ldots & b_{1r} a_{1r} \\ b_{21} a_{21} & b_{22} a_{22} & \ldots & b_{2r} a_{2r} \\ \vdots & \vdots & \ddots & \vdots \\ b_{L1} a_{L1} & b_{L2} a_{L2} & \ldots & b_{Lr} a_{Lr} \end{bmatrix} \quad (2)
$$

where $b_{ij} a_{ij}$ corresponds to the $j$-th expert in the $i$-th layer.

**Masking Rank-1 LoRA Experts.** On the basis of cellular decomposition, we introduce a mask matrix $M \in \mathbb{B}^{L \times r}$, where $\mathbb{B}$ denotes the binary set $\{0, 1\}$, and each element $m_{ij}$ of $M$ is a member of this set. The update matrices across $L$ layers are:

$$
[\Delta W_1 \, \Delta W_2 \, \ldots \, \Delta W_L]^T = M \odot \mathcal{E} \quad (3)
$$

where $\odot$ is the Hadamard product. During the parameter initialization phase, each $a_{ij}$ and $b_{ij}$ is initialized separately to ensure that each expert starts the updating process in a distinct initial state, with $a_{ij}$ being drawn from a random Gaussian distribution and $b_{ij}$ set to zero.

### 3.3 Adaptive Masking Through Expert-Level Dropout

**Adaptive Coefficients.** We hypothesize that the contribution of each expert to the model's performance varies, thus necessitating the allocation of unique weights to each component. In a similar vein, $\Delta W_l = \sum_{i=1}^r \lambda_{li} w_{li} = \sum_{i=1}^r \lambda_{li} b_{li} a_{li}$. So the total update matrices are:

$$
[\Delta W_1 \, \Delta W_2 \, \ldots \, \Delta W_L]^T = M \odot \Lambda \odot \mathcal{E} = M \odot \begin{bmatrix} \lambda_{11} & \lambda_{12} & \ldots & \lambda_{1r} \\ \lambda_{21} & \lambda_{22} & \ldots & \lambda_{2r} \\ \vdots & \vdots & \ddots & \vdots \\ \lambda_{L1} & \lambda_{L2} & \ldots & \lambda_{Lr} \end{bmatrix} \odot \mathcal{E} \quad (4)
$$

where $\Lambda$ is the adaptive coefficient matrix, $\lambda_{ij}$ denotes the weight assigned to the $j$-th expert in the $i$-th layer, and all the elements in $\Lambda$ are initialized to 1 to ensure that all experts contribute equally during the initial phase. Notably, IncreLoRA (Zhang et al., 2023a) has also adopted a similar strategy of scaling coefficients. However, their primary goal is to enable the update matrix to be decomposed in a manner similar to SVD in AdaLoRA (Zhang et al., 2023b), facilitating subsequent incremental allocation. In their approach, the initial values of the scaling coefficients $\lambda$ are set to zero.

**Fixed Masking.** Drawing inspiration from MoLA (Gao et al., 2024), which explores layer-wise expert allocation under the MoE framework, we specifically design fixed masks to investigate which layer–lower, middle, or upper–requires a more dense application of masks, as the five representative masks shown in Fig. 2. When fixed masking is applied, the masked experts are permanently discarded, meaning they do not undergo gradient updates and are not utilized during inference.

**Stochastic Masking.** Similar to the fixed patterns, we have implemented five variations of stochastic masking by employing expert-level dropout to $\Lambda$ and adjusting the dropout rate $p$ across different layers, as illustrated in the bottom of Fig. 2. Accordingly, the total update matrix is given by:

$$[\Delta W_1 \ \Delta W_2 \ \ldots \ \Delta W_L]^T = M \odot \Lambda \odot \mathcal{E} = (\text{Dropout}(\Lambda, p)) \odot \mathcal{E} \quad (5)$$

where $p$ represents the probability of an element being set to zero. During the training process, experts are randomly discarded, and the outputs of the remaining experts are scaled up to maintain the expected output magnitude. Consequently, these discarded experts do not undergo gradient updates during that iteration. During the inference, all experts are activated and used without any scaling.

**Mixed Masking.** Building on the aforementioned fixed masking and stochastic masking, we further propose mixed masking. In this strategy, certain experts are permanently masked, while stochastic masking is applied to the remaining experts. Accordingly, there are also five representative masks.

## 4 EXPERIMENTS

### 4.1 SETUP

**Datasets and Metrics.** We evaluate our method on a total of 24 downstream tasks in two groups following VPT (Jia et al., 2022): (1) VTAB-1k (Zhai et al., 2019) is a large-scale transfer learning benchmark consisting of a collection of 19 vision datasets, which are clustered into three domains: Natural, Specialized, and Structured. The final model used for evaluation is trained using the full 1,000 examples in each dataset. Following Jia et al. (2022), we use top-1 classification accuracy (%) averaged within each domain as the metric; (2) FGVC is a benchmark for fine-grained visual classification tasks including CUB-200-2011 (Wah et al., 2011), NABirds (Van Horn et al., 2015), Oxford Flowers Nilsback & Zisserman (2008), Stanford Dogs (Gebru et al., 2017) and Stanford Cars (Khosla et al., 2011). We follow the validation splits in VPT (Jia et al., 2022). We also report the top-1 accuracy averaged on the FGVC datasets. The average accuracy score on the test set with three runs is reported. More details are in Appendix A.1.

**Pretrained Backbones.** For a fair comparison, we follow SPT (He et al., 2023) and mainly use ViT-B/16 (Dosovitskiy et al., 2020) pre-trained on supervised ImageNet-21K (Deng et al., 2009) as our default backbone. We only implement our proposed MLAE in the Q-K-V linear transformation layer across the 12 blocks of ViT-B, and this alone enables us to surpass GLoRA (Chavan et al., 2023) which fine-tunes all linear layers in the model. However, MLAE can be implemented in any linear layer across various network architectures.

**Implementation.** All the experiments are trained by use of PyTorch on an NVIDIA GeForce RTX 3090 GPU. Following SPT (He et al., 2023), we employ the AdamW optimizer (Loshchilov & Hutter, 2018) with cosine learning rate decay in the training process. We set the batch size, learning rate, and weight decay as 64, 5e-4, 1e-4, respectively. Based on previous works (He et al., 2023; Jia et al., 2022; Zhang et al., 2022), we also follow the same standard data augmentation pipeline, e.g., processing the image with randomly resize crop to $224 \times 224$ and a random horizontal flip for the FGVC datasets. Additionally, during data preprocessing, we use ImageNet inception mean and standard deviation for normalization. The number of training epochs is set to 500 for VTAB-1k and 300 for FGVC to achieve better convergence. For the coefficient initialization and dropout probability choices, we provide more details in Appendix A.2.

### 4.2 RESULTS

Our proposed approach achieves SOTA performance on both VTAB-1k and FGVC benchmarks.

**Performance on VTAB-1k.** We first compare MLAE with the state-of-the-art PEFT methods on ViT, as reported in Table 1. We initially note that all PEFT methods significantly surpass full

Table 1: **Full results on VTAB-1K benchmark**. "# param" specifies the total number of trainable parameters in backbones. Average accuracy and # param are averaged over group-wise mean values. The best result is in **bold**, and the second-best result is underlined.

| | # param (M) | Natural | | | | | | | Specialized | | | | Structured | | | | | | | | Average |
| --- | --- | --- | --- | --- | --- | --- | --- | --- | --- | --- | --- | --- | --- | --- | --- | --- | --- | --- | --- | --- | --- |
| | | Cifar100 | Caltech101 | DTD | Flower102 | Pets | SVHN | Sun397 | Camelyon | EuroSAT | Resisc45 | Retinopathy | Clevr-Count | Clevr-Dist | DMLab | KITTI-Dist | dSpr-Loc | dSpr-Ori | sNORB-Azim | sNORB-Ele | |
| *Traditional Finetuning* | | | | | | | | | | | | | | | | | | | | | |
| Full | 85.8 | 68.9 | 87.7 | 64.3 | 97.2 | 86.9 | 87.4 | 38.8 | 79.7 | 95.7 | 84.2 | 73.9 | 56.3 | 58.6 | 41.7 | 65.5 | 57.5 | 46.7 | 25.7 | 29.1 | 68.9 |
| Linear | 0 | 64.4 | 85.0 | 63.2 | 97.0 | 86.3 | 36.6 | 51.0 | 78.5 | 87.5 | 68.5 | 74.0 | 34.3 | 30.6 | 33.2 | 55.4 | 12.5 | 20.0 | 9.6 | 19.2 | 57.6 |
| *PEFT methods* | | | | | | | | | | | | | | | | | | | | | |
| VPT-Deep | 0.53 | **78.8** | 90.8 | 65.8 | 98.0 | 88.3 | 78.1 | 49.6 | 81.8 | 96.1 | 83.4 | 68.4 | 68.5 | 60.0 | 46.5 | 72.8 | 73.6 | 47.9 | 32.9 | 37.8 | 72.0 |
| Adapter | 0.16 | 69.2 | 90.1 | 68.0 | 98.8 | 89.9 | 82.8 | 54.3 | 84.0 | 94.9 | 81.9 | 75.5 | 80.9 | 65.3 | 48.6 | 78.3 | 74.8 | 48.5 | 29.9 | 41.6 | 73.9 |
| AdaptFormer | 0.16 | 70.8 | 91.2 | 70.5 | 99.1 | 90.9 | 86.6 | 54.8 | 83.0 | 95.8 | 84.4 | 76.3 | 81.9 | 64.3 | 49.3 | 80.3 | 76.3 | 45.7 | 31.7 | 41.1 | 74.7 |
| LoRA | 0.29 | 67.1 | 91.4 | 69.4 | 98.8 | 90.4 | 85.3 | 54.0 | 84.9 | 95.3 | 84.4 | 73.6 | 82.9 | **69.2** | 49.8 | 78.5 | 75.7 | 47.1 | 31.0 | 44.0 | 74.5 |
| NOAH | 0.36 | 69.6 | 92.7 | 70.2 | 99.1 | 90.4 | 86.1 | 53.7 | 84.4 | 95.4 | 83.9 | 75.8 | 82.8 | 68.9 | 49.9 | 81.7 | 81.8 | 48.3 | 32.8 | 44.2 | 75.5 |
| FacT | 0.07 | 70.6 | 90.6 | 70.8 | 99.1 | 90.7 | 88.6 | 54.1 | 84.8 | 96.2 | 84.5 | 75.7 | 82.6 | 68.2 | 49.8 | 80.7 | 80.8 | 47.4 | 33.2 | 43.0 | 75.6 |
| SSF | 0.24 | 69.0 | 92.6 | 75.1 | 99.4 | 91.8 | 90.2 | 52.9 | 87.4 | 95.9 | 87.4 | 75.5 | 75.9 | 62.3 | 53.3 | 80.6 | 77.3 | 54.9 | 29.5 | 37.9 | 75.7 |
| RepAdapter | 0.22 | 72.4 | 91.6 | 71.0 | 99.2 | 91.4 | **90.7** | 55.1 | 85.3 | 95.9 | 84.6 | 75.9 | 82.3 | 68.0 | 50.4 | 79.9 | 80.4 | 49.2 | **38.6** | 41.0 | 76.1 |
| SPT-Adapter | 0.38 | 72.9 | 93.2 | 72.5 | 99.3 | 91.4 | 88.8 | 55.8 | 86.2 | 96.1 | 85.5 | 75.5 | 83.0 | 68.0 | 51.9 | 81.2 | 82.4 | 51.9 | 31.7 | 41.2 | 76.4 |
| SPT-LoRA | 0.54 | 73.5 | 93.3 | 72.5 | 99.3 | 91.5 | 87.9 | 55.5 | 85.7 | 96.2 | 85.9 | 75.9 | **84.4** | 67.6 | 52.5 | 82.0 | 81.0 | 51.1 | 30.2 | 41.3 | 76.4 |
| GLoRA | 0.29 | 76.1 | 92.7 | **75.3** | **99.6** | 92.4 | 90.5 | 57.2 | 87.5 | 96.7 | 88.1 | 76.1 | 81.0 | 66.2 | 52.4 | 84.9 | 81.8 | 53.3 | 33.3 | 39.8 | 77.3 |
| GLoRA | 0.86 | 76.4 | 92.9 | 74.6 | **99.6** | 92.5 | 90.5 | 57.8 | 87.3 | **96.8** | 88.0 | 76.0 | 83.1 | 67.3 | 54.5 | **86.2** | 83.8 | 52.9 | 37.0 | 41.4 | 78.0 |
| **MLAE (Ours)** | 0.30 | 77.8 | 94.7 | 74.8 | 99.4 | **92.6** | 90.6 | **58.4** | **88.4** | 96.5 | **88.5** | **76.7** | 84.3 | 67.2 | **55.7** | 82.6 | **87.8** | **57.1** | 35.7 | **47.7** | **78.8** |

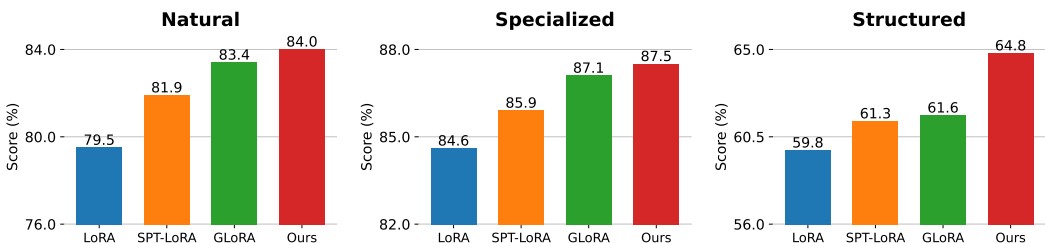

Figure 3: **Domain-wise average scores on VTAB-1k benchmark.** Our MLAE performs best in all three domains, especially in Structured domain, surpassing GLoRA by 3.2% under a similar budget.

finetuning in performance, whereas linear probing, which involves merely adjusting the classifier, yields substantially poorer results. Compared to these methods, we can see that under a similar budget (e.g., 0.29M trainable parameters for GLoRA (Chavan et al., 2023) vs. 0.30M for MLAE), MLAE performs better on VTAB-1k. In particular, by slightly adjusting the dropout rates across different datasets, MLAE can achieve SOTA performance with an average accuracy score of 78.8%, even better than GLoRA with almost tripled budget (78.0%). MLAE achieves the best performance in 9 out of 19 datasets in the VTAB-1k benchmark and secures the second-best results in 7 additional datasets. Moreover, intuitively from Fig. 3, we can see that compared to the previous LoRA-based PEFT methods, MLAE performs the best in all three domains, especially in the Structured domain, surpassing the previous SOTA method GLoRA by 3.2% under a comparable budget. Notably, SPT (He et al., 2023) reveals that, by measuring the maximum mean discrepancy between the source and target domains, Structured datasets have larger domain gaps from the pre-training source (Deng et al., 2009) compared to Natural and Specialized datasets. The significant improvement in the Structured domain indicates that MLAE may contribute to mitigating domain gaps by compelling each expert to extract more diverse information from images.

**Performance on FGVC.** To validate the effectiveness of our method in fine-grained visual classification tasks, we conduct experiments on the FGVC benchmark using three random seeds. The experimental results indicate that MLAE achieves the best performance across all seeds, demonstrating its stability in exceptional performance and parameter efficiency. As shown in Table 2, the traditional fine-tuning method FULL, despite using 100% of the parameters, only achieves an accuracy of 88.5%. In contrast, our method achieves an accuracy of 90.9% with only 0.34% trainable parameters. Compared to other PEFT methods, Adapter (85.7%), LoRA-8 (86.0%), SSF (90.7%), and SPT-LoRA (90.1%), MLAE achieves the highest accuracy with the least tuned parameters. These

Table 2: **Results on FGVC benchmark.** "Tuned/Total" denotes the fraction of trainable parameters.

| ViT-B/16 | FGVC | |
|---|---|---|
| | Tuned/Total | Mean Acc. |
| *Traditional Finetuing* | | |
| FULL | 100% | 88.5 |
| *PEFT methods* | | |
| Adapter | 0.41% | 85.7 |
| LoRA-8 | 0.55% | 86.0 |
| SSF | 0.39% | 90.7 |
| VPT-Deep | 0.98% | 89.1 |
| SPT-LoRA | 0.60% | 90.1 |
| MLAE (Ours) | 0.34% | **90.9** |

Table 3: **Inference efficiency comparison of MLAE with existing methods during inference.** $\Delta P$ and $\Delta F$ denote the additional parameters and FLOPs by PEFT methods, respectively. The inference throughput is defined as images per second (imgs/sec).

| Method | $\Delta P$ (M) (↑) | $\Delta F$ (G) (↑) | Throughput (imgs/sec) | | |
|---|---|---|---|---|---|
| | | | bs=1 | bs=4 | bs=16 |
| Full tuning | 0 | 0 | 91.5 | 375.7 | 539.5 |
| VPT | 0.55 | 5.6 | 86.1 | 283.5 | 381.5 |
| Adapter | 0.16 | 0.03 | 70.9 | 306.6 | 504.7 |
| AdaptFormer | 0.16 | 0.03 | 71.4 | 309.9 | 508.1 |
| NOAH | 0.12 | 0.02 | 72.1 | 312.7 | 492.9 |
| LoRA | | | 91.5 | 375.7 | 539.6 |
| GLoRA | 0 | 0 | 91.5 | 375.7 | 539.6 |
| MLAE (Ours) | | | 91.5 | 375.2 | 538.5 |

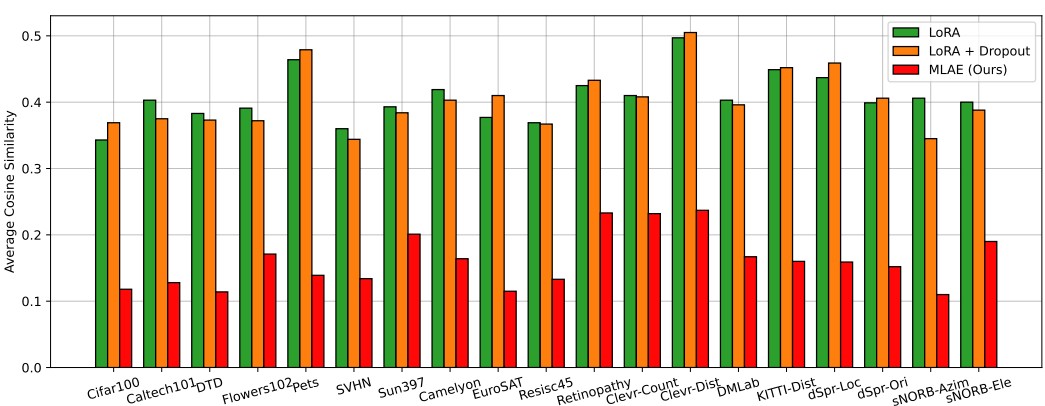

Figure 4: Comparisons of average parameter similarity between MLAE and LoRA baselines.

results highlight our method's capability to achieve optimal performance with minimal trainable parameters, proving the effectiveness and superiority of MLAE in fine-grained visual classification tasks. More detailed experimental results can be found in Appendix A.3.

**Efficiency Analysis.** Following RepAdapter (Luo et al., 2023), we present the final inference throughput of various PEFT methods in Table 3, all computed on an NVIDIA RTX 3090 GPU. In the table, we employ ViT-B/16 as the vision model. Our method, which builds upon LoRA, maintains the same count of additional parameters, FLOPs, and throughput as LoRA (i.e., ours: 538.5 imgs/sec vs. 539.6 imgs/sec for LoRA). However, MLAE, due to the introduction of dropout, requires a slightly longer training time compared to a single run of LoRA — averaging 40 minutes per VTAB-1k task, whereas LoRA averages 26 minutes. The GPU memory consumption for MLAE is 9.8 GB, compared to 9 GB for LoRA and 13 GB for GLoRA.

### 4.3 ANALYSIS AND VISUALIZATIONS

To gain deeper insights into the underlying mechanisms of MLAE, we first provide a theoretical intuition explaining why expert-level dropout leads to diversified experts and how reducing parameter similarity contributes to performance gains. Second, we conduct analyses and visualizations from the perspectives of parameter similarity and feature attention maps to substantiate our findings. The results indicate that MLAE significantly reduces parameter similarity compared to vanilla LoRA, allowing each expert to extract more diverse feature information, ultimately enhancing model performance.

**Theoretical Intuition.** Previous work (Gal & Ghahramani, 2016; Maeda, 2014) has theoretically demonstrated that Dropout can be interpreted as an approximation to a Bayesian posterior over network weights. This Bayesian perspective suggests that Dropout facilitates averaging predictions across a distribution of various network architectures. By randomly omitting neurons during train-

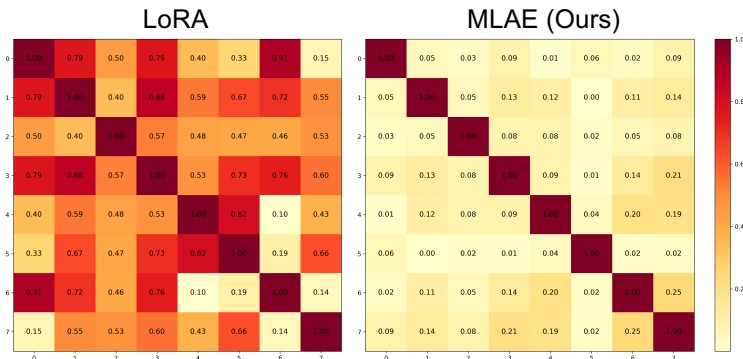

Figure 5: Confusion matrix comparison of overall parameter similarity on Cifar100 ($r = 8$).

ing, Dropout effectively emulates sampling from a distribution of different network subsets. This introduces uncertainty into the model, enabling it to train across multiple sub-networks (Lakshmi-narayanan et al., 2017). Specifically, we employ different random initializations for various experts; due to unique initial conditions and stochasticity in the training process, each expert can converge to a different basin of the posterior, thereby exhibiting diversity. Additionally, Grewal & Bui (2021); Nam et al. (2021) enhanced the performance of Bayesian Model Averaging (BMA) (Gal & Ghahramani, 2016) by increasing the diversity among ensemble members. Given that Dropout is a mathematical equivalent form of BMA, reducing parameter similarity (under rank-1 decomposition) consequently implies increased diversity among experts which can finally lead to performance gains.

**Parameter Cosine Similarity Analysis.** We first explore the differences in cosine similarity among parameters within the incremental matrices learned by LoRA and our proposed MLAE. The calculation details are provided in Appendix A.4. Fig. 4 displays the average cosine similarity among model parameters for 19 visual tasks on the VTAB-1k benchmark across three different settings. Fig. 5 shows the overall parameter similarity confusion matrices on CIFAR-100, comparing LoRA and MLAE (Ours), with MLAE exhibiting significantly lower parameter similarity. We can observe from the Fig. 4: (i) Applying dropout directly on $\Delta W$, i.e., LoRA+Dropout, does not always effectively reduce parameter similarity. This interesting phenomenon suggests that directly applying dropout to the incremental matrix may impact the learning of model parameters due to its randomness, resulting in more uniform parameter updates on certain datasets and thus increasing similarity; (ii) Our method MLAE exhibits lower average cosine similarity across all datasets. Unlike LoRA+Dropout, MLAE performs expert-level dropout by stochastically masking certain experts during the training process. Results from Table 1 and Fig. 4 demonstrate that MLAE can effectively and significantly reduce parameter similarity, thereby achieving higher model performance.

**Feature Attention-map Visualization.** To visualize the feature representations learned by each expert in MLAE, focusing on a single block, we show a few representative feature attention maps of the same expert across different samples in Fig. 6. From a horizontal perspective, there are notable variations among experts within the same block. For instance, with the first sunflower, some experts mainly focus on the petals (*Expert 2, 3*), while others concentrate more on the center of the flower (*Expert 5*). Vertically, we notice consistency among experts when focusing on the same class. For example, *Expert 1, 8* focus on the sunflower's petals, while *Expert 4* consistently focuses on the background, etc. When the image switches to another flower (e.g., Cosmos), the attention distribution changes. While some experts still focus on the petals and background, others shift to the center (*Expert 3*) or the background (*Expert 7*). This demonstrates that the experts in MLAE exhibit a distinct degree of specialization but also can capture a broad spectrum of feature representations, highlighting the diversity and complementarity of the expert modules in feature extraction and representation. We speculate that such differentiated focus areas and consistency help the model obtain more comprehensive and diverse feature representations when dealing with complex tasks, thereby enhancing the overall performance and generalization ability of the model.

### 4.4 MASKING STRATEGY COMPARISONS

We conduct an in-depth investigation of masking strategies with different configurations for MLAE. **Configurations.** Recent work such as MoLA (Gao et al., 2024) manually sets the number of LoRA

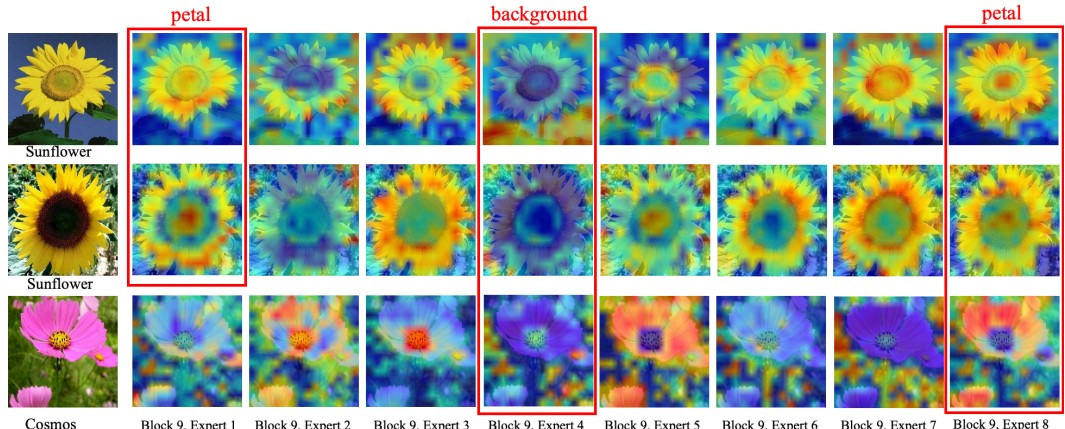

Figure 6: **Visualization of attention patterns within Block 9 of Flowers102 model across different samples.** More examples of other dataset-specific model can also be found in Appendix A.7.

Table 4: Results of three masking strategies under different configurations illustrated in Fig. 2 (all with the same budget of total rank = 96).

| Masking strategy | Incremental | Hourglass | Decremental | Protruding | Random | Ours |
|---|---|---|---|---|---|---|
| fixed | 76.8 | 75.5 | 75.7 | 76.1 | 76.5 | |
| stochastic | 77.7 | 76.3 | 76.0 | 77.6 | — | **78.8** |
| mixed | 78.2 | 77.5 | 77.7 | 78.0 | 78.1 | |

experts for different layers. We explore a similar implementation within our MLAE, by assigning varying experts to layers based on our decomposition with ***fixed masking***. Compared to allocating the same number of experts to all layers, this is equivalent to permanently masking some experts in a fixed manner. We also explore a novel ***stochastic masking*** approach, allocating an identical number of experts across all layers and applying expert-level masks through stochastic masking during training. The expert-level mask refers to setting an expert's weight coefficient to zero with a certain probability, thus masking the expert during a specific iteration. To achieve this, we use dropout on the coefficient matrix, fixing the number of experts per layer and adjusting the dropout probability across layers. Combining the two aforementioned masking patterns, we explore a third masking strategy, i.e., ***mixed masking***, using the same dropout probability across all layers while maintaining the layer-wise allocation of experts as in fixed masking. We choose to use the same dropout probability for all layers based on findings from the stochastic masking that a consistent dropout rate yields optimal results. The purpose of this investigation is to figure out whether expert-level stochastic masking can bring performance gains to fixed masking and to verify our method's universality and efficacy. The configurations are depicted in Fig. 2, and implementation details are provided in Appendix A.5.

**Results.** All experiments with the above settings are conducted under an identical budget. From the results summarized in Table 4, we have the following findings: (i) The configuration with a uniform dropout rate applied across all layers through stochastic masking (Ours) outperforms the other configurations, achieving a score of 78.8%. (ii) Stochastic and mixed masking significantly outperforms fixed masking, indicating that dynamic expert-level masking is preferable. The experimental outcomes validate the application of stochastic masking strategy, demonstrating its ability to facilitate experts in learning diverse feature information, thereby improving model performance.

## 4.5 ABLATION STUDY

**Key Components.** To thoroughly evaluate the proposed MLAE, we conduct extensive ablation studies. In Table 5, we evaluate the impact of three key components on model performance. Results show that when using cellular decomposition alone, the model achieves an average accuracy of only 76.6%. However, the performance significantly improves by 2.0% when both cellular decomposition and expert-level masking techniques are applied. Employing all three techniques together elevates the model's performance to a new SOTA accuracy of 78.8%. This indicates that while cellular decom-

Table 5: Ablations on key components in MLAE.

| Decomposition | Mask | Adaptive | VTAB-1k | | | |
| --- | --- | --- | --- | --- | --- | --- |
| | | | Natural | Specialized | Structured | Mean Acc. |
| ✓ | ✗ | ✗ | 80.4 | 86.1 | 63.4 | 76.6 |
| ✓ | ✗ | ✓ | 80.5 | 86.0 | 63.2 | 76.6 |
| ✓ | ✓ | ✗ | 83.9 | 87.4 | 64.6 | 78.6 |
| ✓ | ✓ | ✓ | **84.0** | **87.5** | **64.8** | **78.8** |

Table 6: Rank-1, -2, -4 Submatrix.

| Submatrix Rank | Mean Acc. |
| --- | --- |
| $\underbrace{4\ 4}_{2\times4}$ | 78.2 |
| $\underbrace{2\ 2\ 2\ 2}_{4\times2}$ | 78.3 |
| $\underbrace{1\ 1\ 1\ 1\ 1\ 1\ 1\ 1}_{8\times1}$ | **78.8** |

Table 7: Total Budget.

| #Experts | Mean Acc. |
| --- | --- |
| $\underbrace{1\ 1}_{2\times1}$ | 77.3 |
| $\underbrace{1\ 1\ 1\ 1}_{4\times1}$ | 78.2 |
| $\underbrace{1\ 1\ 1\ 1\ 1\ 1\ 1\ 1}_{8\times1}$ | **78.8** |

Table 8: Initialization.

| Coeff. | Mean Acc. |
| --- | --- |
| 0.125 | 78.0 |
| 0.25 | 78.2 |
| 0.5 | 78.2 |
| 1 | 78.3 |
| 2 | 78.1 |
| 4 | 77.9 |

Table 9: Probability.

| $p$ | Mean Acc. |
| --- | --- |
| 0 | 76.6 |
| 0.1 | 77.1 |
| 0.3 | 77.3 |
| 0.5 | 77.5 |
| 0.7 | 77.3 |
| 0.9 | 74.4 |

position serves as a solid foundation, expert-level masking can significantly enhance performance, and the adaptive coefficients technique further optimizes model behavior as a complement.

**Submatrix Rank and Budget.** While fine-tuning the incremental matrix $\Delta W$ under the same budget of trainable parameters, we assess the effect of different expert rank configurations. The results in Table 6 indicate that, rank-1 submatrix configuration achieves the highest performance. Next, we explore the impact of tuning $\Delta W$ using different numbers of rank-1 experts in Table 7. The results indicate that increasing the number of rank-1 experts can significantly improve model performance, possibly because more experts can process diverse features more finely, thus enhancing the overall performance. However, it is worth noting that indiscriminately increasing the number of experts may not continuously yield performance gains and instead can increase computational demands and resource consumption. Therefore, choosing an appropriate number of experts to balance performance improvement and resource consumption is crucial.

**Hyperparameters of Initialization and Probability.** Finally, inspired by RepAdapter (Luo et al., 2023) for searching the hyperparameter scaling coefficient, we also explore the initialization values of the coefficient matrices and the dropout probability in our method in Table 8 and Table 9. The first observation is that an initialization value of 1 for the coefficient matrices yields slightly better performance while a probability of 0.5 achieves the best performance on the VTAB-1k benchmark. However, it is noteworthy that the optimal values for the initialization and probability may vary across different datasets to achieve peak performance. More detailed results can be found in Appendix A.6.

## 5 CONCLUSION AND DISCUSSION

In this paper, we introduce masking strategies into visual PEFT for the first time to enhance learning diversity and parameter quality, proposing a novel method called MLAE. We conducted extensive research on various masking strategies, finding that using dropout to generate masks for our adaptive coefficients matrix yields the best performance and enables the model to learn more diverse knowledge. We carried out exhaustive experiments on 24 datasets across VTAB-1k and FGVC benchmarks, demonstrating the reliability and superiority of our method.

**Limitations.** Compared to other methods, a drawback of MLAE is the variability in the probability of stochastic masking across different datasets. Hence, it is essential to search for the optimal probability $p$, with the search range and detailed results provided in the appendix A.2.

**Future Works.** This shortcoming points us towards further research directions: Can we develop a metric to determine the probability of random masking based on dataset characteristics or training performance? Additionally, how can we find the optimal $p$ for different layers within the model? These questions are intriguing and warrant deeper exploration.

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

# A   APPENDIX

## A.1   DATASET DETAILS

**VTAB-1k** (Zhai et al., 2019) is a large-scale transfer learning benchmark consisting of a collection of 19 vision datasets, which are clustered into three domains: Natural, Specialized, and Structured. The Natural domain contains natural images that are captured by standard cameras. The Specialized domain contains images captured by specialist equipment for remote sensing and medical purposes. The Structured domain is designed specifically for scene structure understanding, such as object counting, depth prediction, and orientation prediction. Each dataset contains 800 training and 200 validation samples, while the size of the original test set varies. Each task only contains 1,000 training samples, thus is extremely challenging.

**FGVC** is a benchmark for fine-grained visual classification tasks including CUB-200-2011 (Wah et al., 2011), NABirds (Van Horn et al., 2015), Oxford Flowers (Nilsback & Zisserman, 2008), Stanford Dogs (Gebru et al., 2017) and Stanford Cars (Khosla et al., 2011). Each FGVC dataset contains between 55 to 200 classes and a few thousand images for train, validation, and test. We follow the validation splits in VPT (Jia et al., 2022) for there are no public train/val splits in these datasets.

## A.2   IMPLEMENTATION DETAILS

Inspired by RepAdapter (Luo et al., 2023) for searching the hyperparameter scaling coefficient, we also explore the initialization values of the coefficient matrices and the dropout probability in our method. Dropout probability is searched from [0.0, 0.1, 0.3, 0.5, 0.7, 0.9] and the initialization value of expert coefficient is searched from [0.125, 0.25, 0.5, 1, 2, 4]. The detailed dataset-specific optimal values are shown in Table 10.

Table 10: Optimal values for initialization and probability on the VTAB-1K benchmark.

| Dataset | Cifar100 | Caltech101 | DTD | Flowers102 | Pets | SVHN | Sun397 | Camelyon | EuroSAT | Resisc45 | Retinopathy | Clevr-Count | Clevr-Dist | DMLab | KITTI-Dist | dSpr-Loc | dSpr-Ori | sNORB-Azim | sNORB-Ele |
|---|---|---|---|---|---|---|---|---|---|---|---|---|---|---|---|---|---|---|---|
| Dropout probability | 0.7 | 0.1 | 0.7 | 0.7 | 0.7 | 0.5 | 0.7 | 0.7 | 0.5 | 0.5 | 0.7 | 0 | 0 | 0.5 | 0.5 | 0.5 | 0.7 | 0 | 0 |
| Coeff Initialization | 2 | 0.5 | 0.25 | 1 | 2 | 4 | 4 | 1 | 4 | 2 | 0.25 | 0.125 | 0.125 | 2 | 1 | 1 | 4 | 1 | 0.25 |

## A.3   DETAILED RESULTS ON THE FGVC BENCHMARK

Table 11: Performance comparisons on five FGVC datasets with ViT-B/16 models pre-trained on ImageNet-21K.

| Method \ Dataset | CUB-200-2011 | NABirds | Oxford Flowers | Stanford Dogs | Stanford Cars | Mean | Params. (M) |
|---|---|---|---|---|---|---|---|
| Full fine-tuning | 87.3 | 82.7 | 98.8 | 89.4 | 84.5 | 88.54 | 85.98 |
| Linear probing | 85.3 | 75.9 | 97.9 | 86.2 | 51.3 | 79.32 | 0.18 |
| Adapter | 87.1 | 84.3 | 98.5 | 89.8 | 68.6 | 85.67 | 0.41 |
| Bias | 88.4 | 84.2 | 98.8 | 91.2 | 79.4 | 88.41 | 0.28 |
| VPT-Shallow | 86.7 | 78.8 | 98.4 | 90.7 | 68.7 | 84.62 | 0.25 |
| VPT-Deep | 88.5 | 84.2 | 99.0 | 90.2 | 83.6 | 89.11 | 0.85 |
| SPT-LoRA | 88.6 | 83.4 | 99.5 | 91.4 | 87.4 | 90.06 | 0.69 |
| SSF | 89.5 | **85.7** | **99.6** | 89.6 | **89.2** | 90.72 | 0.39 |
| MLAE (seed 0) | 89.6 | 84.5 | 99.2 | 91.6 | 89.0 | 90.78 | 0.29 |
| MLAE (seed 1) | 89.6 | 85.1 | 99.2 | 91.7 | 89.1 | 90.94 | 0.29 |
| MLAE (seed 42) | 89.6 | 85.3 | 99.3 | 91.5 | 88.5 | 90.84 | 0.29 |
| MLAE (Avg) | **89.6** | 85 | 99.2 | **91.6** | 88.9 | **90.86** | 0.29 |

### A.4 CALCULATION OF COSINE SIMILARITY AMONG PARAMETERS

Here, we briefly explain the calculation of cosine similarity among parameters. Since MLAE decomposes the incremental matrix $\Delta W$ into $r$ parameter matrices of size $768 \times 2304$, we can directly flatten each expert's parameter matrix into a one-dimensional tensor, then calculate the cosine similarity between each pair of tensors, and finally average the similarity values within all 12 blocks of the model to represent the overall model's cosine similarity. When calculating the parameter similarity for $\Delta W$ learned by LoRA, we perform a special operation by extracting a column from matrix $B$ ($B \in \mathbb{R}^{d_{in} \times r}$) and a row from matrix $A$ ($A \in \mathbb{R}^{r \times d_{out}}$) for matrix multiplication, resulting in a parameter submatrix of size $768 \times 2304$ (the number of rows and columns depends on the set value of $r$). We can thus obtain $r$ parameter submatrices, and the subsequent processing is the same as for MLAE.

### A.5 CONFIGURATIONS OF MASKING STRATEGIES

**Detailed Configurations of Fixed Masking.** The *incremental* pattern setting assigns the number of experts within each block in ViT-B from block 1 to block 12 as [2, 2, 2, 6, 6, 6, 10, 10, 10, 14, 14, 14]. Similarly, the *decremental* pattern setting is set as [14, 14, 14, 10, 10, 10, 6, 6, 6, 2, 2, 2], the *hourglass* pattern setting uses [14, 14, 14, 2, 2, 2, 2, 2, 2, 14, 14, 14], and the *protruding* pattern setting uses [2, 2, 2, 14, 14, 14, 14, 14, 2, 2, 2]. In the *random* pattern setting, the number of experts per layer is randomly assigned an integer value within the range of 1 to 14, with the constraint that the average value across all layers is 8. All these settings are under the same trainable parameter budget.

**Detailed Configurations of Stochastic Masking.** The number of experts across different layers is fixed at 8 for a fair comparison. Note that the names of these patterns are based on the expected number of remaining experts, not the dropout rates, with the assumption that higher dropout probabilities generally result in fewer experts remaining active in a layer. For example, a layer with a dropout rate of 0.8 typically retains only 2-3 experts during training, while a dropout rate of 0.0 means that all experts are retained. Hence, the dropout probability assignment from block 1 to block 12 of the *incremental* pattern is set as [0.8, 0.8, 0.7, 0.7, 0.6, 0.6, 0.5, 0.4, 0.3, 0.2, 0.1, 0.0]. Similarly, the *decremental* pattern setting is set as [0.0, 0.1, 0.2, 0.3, 0.4, 0.5, 0.6, 0.6, 0.7, 0.7, 0.8, 0.8], the *hourglass* pattern setting uses [0.0, 0.1, 0.3, 0.5, 0.6, 0.8, 0.8, 0.6, 0.5, 0.3, 0.1, 0.0], the *protruding* pattern setting is [0.8, 0.6, 0.5, 0.3, 0.1, 0.0, 0.0, 0.1, 0.3, 0.5, 0.6, 0.8], and a uniform dropout rate across all layers (**Ours**).

### A.6 EXPERIMENTAL RESULTS DETAILS

The following tables provide detailed results of our ablation experiments on the VTAB-1K benchmark. Table 12 compares different rank submatrices, Table 13 examines the impact of varying budget allocations, Table 14 details the results of different initialization value for expert coefficient, and Table 15 explores the effect of expert-level masking probability across layers. Each table includes performance metrics across Natural, Specialized, and Structured datasets, with averages provided for comprehensive evaluation.

Table 12: Detailed results of different rank submatrix on VTAB-1K benchmark variants.

| Sub-matrix Rank | Natural | | | | | | | Specialized | | | | Structured | | | | | | | | Average |
|---|---|---|---|---|---|---|---|---|---|---|---|---|---|---|---|---|---|---|---|---|
| | Cifar100 | Caltech101 | DTD | Flower102 | Pets | SVHN | Sun397 | CamelLyon | EuroSAT | Resisc45 | Retinopathy | Clevr-Count | Clevr-Dist | DMLab | KITTI-Dist | dSpr-Loc | dSpr-Ori | sNORB-Azim | sNORB-Ele | |
| $\underbrace{4\ 4}_{2\times4}$ | 77.0 | 94.4 | 74.5 | 99.3 | 92.5 | 89.8 | 58.1 | 87.3 | 96.6 | 88.1 | 76.5 | 84.4 | 67.0 | 55.5 | 82.1 | 86.2 | 56.2 | 33.7 | 46.1 | 78.2 |
| $\underbrace{2\ 2\ 2\ 2}_{4\times2}$ | 77.7 | 94.6 | 74.5 | 99.4 | 92.5 | 88.9 | 58.2 | 88.4 | 96.3 | 88.4 | 76.5 | 82.6 | 65.3 | 54.4 | 82.6 | 87.8 | 55.8 | 35.7 | 47.3 | 78.3 |
| $\underbrace{1\ 1\ 1\ 1\ 1\ 1\ 1\ 1}_{8\times1}$ | 77.8 | 94.7 | 74.8 | 99.4 | 92.6 | 90.6 | 58.4 | 88.4 | 96.5 | 88.5 | 76.7 | 84.3 | 67.5 | 55.7 | 82.6 | 87.8 | 57.1 | 35.7 | 47.7 | 78.8 |

Table 13: Detailed results of different budgets on VTAB-1K benchmark.

| # Experts | Natural | | | | | | | Specialized | | | | Structured | | | | | | | | Average |
|---|---|---|---|---|---|---|---|---|---|---|---|---|---|---|---|---|---|---|---|---|
| | Cifar100 | Caltech101 | DTD | Flower102 | Pets | SVHN | Sun397 | Camelyon | EuroSAT | Resisc45 | Retinopathy | Clevr-Count | Clevr-Dist | DMLab | KITTI-Dist | dSpr-Loc | dSpr-Ori | sNORB-Azim | sNORB-Ele | |
| 1 1 (2×1) | 75.5 | 92.3 | 73.7 | 99.3 | 92.1 | 87.9 | 58.2 | 86.8 | 95.9 | 87.7 | 75.8 | 83.4 | 66.0 | 51.4 | 80.2 | 82.5 | 56.7 | 34.7 | 47.1 | 77.3 |
| 1 1 1 1 (4×1) | 76.7 | 94.2 | 74.4 | 99.3 | 92.4 | 89.6 | 58.1 | 87.4 | 96.1 | 88.5 | 76.1 | 83.8 | 68.3 | 53.4 | 80.7 | 86.9 | 56.5 | 35.4 | 47.4 | 78.2 |
| 1 1 1 1 1 1 1 1 (8×1) | 77.8 | 94.7 | 74.8 | 99.4 | 92.6 | 90.6 | 58.4 | 88.4 | 96.5 | 88.5 | 76.7 | 84.3 | 67.5 | 55.7 | 82.6 | 87.8 | 57.1 | 35.7 | 47.7 | 78.8 |

Table 14: Detailed results of different initializations on VTAB-1K benchmark.

| Initialization | Natural | | | | | | | Specialized | | | | Structured | | | | | | | | Average |
|---|---|---|---|---|---|---|---|---|---|---|---|---|---|---|---|---|---|---|---|---|
| | Cifar100 | Caltech101 | DTD | Flower102 | Pets | SVHN | Sun397 | Camelyon | EuroSAT | Resisc45 | Retinopathy | Clevr-Count | Clevr-Dist | DMLab | KITTI-Dist | dSpr-Loc | dSpr-Ori | sNORB-Azim | sNORB-Ele | |
| 0.125 | 76.9 | 94.7 | 74.5 | 99.3 | 92.3 | 87.6 | 58.1 | 87.7 | 96.2 | 87.8 | 76.3 | 84.3 | 67.2 | 52.9 | 81.0 | 87.4 | 55.8 | 34.0 | 47.7 | 78.0 |
| 0.25 | 77.2 | 94.4 | 74.8 | 99.4 | 92.4 | 87.8 | 58.1 | 88.1 | 96.2 | 88.1 | 76.7 | 84.0 | 65.5 | 53.6 | 81.3 | 87.1 | 55.8 | 35.2 | 47.7 | 78.2 |
| 0.5 | 77.5 | 94.7 | 74.5 | 99.4 | 92.6 | 88.6 | 58.1 | 87.9 | 96.2 | 88.1 | 75.7 | 83.6 | 66.3 | 54.6 | 82.0 | 87.7 | 55.7 | 35.3 | 47.4 | 78.2 |
| 1 | 77.7 | 94.6 | 74.5 | 99.4 | 92.5 | 88.9 | 58.2 | 88.4 | 96.3 | 88.4 | 76.5 | 82.6 | 65.3 | 54.4 | 82.7 | 87.8 | 55.8 | 35.7 | 47.3 | 78.3 |
| 2 | 77.8 | 94.3 | 74.3 | 99.4 | 92.6 | 89.1 | 58.2 | 88.1 | 96.3 | 88.5 | 75.7 | 81.4 | 64.1 | 55.7 | 82.0 | 87.7 | 56.8 | 35.3 | 46.0 | 78.1 |
| 4 | 77.0 | 93.6 | 73.8 | 99.4 | 92.5 | 90.6 | 58.4 | 87.2 | 96.5 | 88.5 | 75.3 | 80.7 | 64.2 | 55.5 | 81.9 | 87.1 | 57.1 | 34.8 | 44.7 | 77.9 |

Table 15: Detailed results of probability on VTAB-1K benchmark (without the best initialization).

| $p$ | Natural | | | | | | | Specialized | | | | Structured | | | | | | | | Average |
|---|---|---|---|---|---|---|---|---|---|---|---|---|---|---|---|---|---|---|---|---|
| | Cifar100 | Caltech101 | DTD | Flower102 | Pets | SVHN | Sun397 | Camelyon | EuroSAT | Resisc45 | Retinopathy | Clevr-Count | Clevr-Dist | DMLab | KITTI-Dist | dSpr-Loc | dSpr-Ori | sNORB-Azim | sNORB-Ele | |
| 0 | 70.7 | 94.9 | 70.2 | 99.1 | 90.1 | 88.0 | 53.0 | 88.0 | 95.9 | 86.3 | 73.6 | 82.3 | 64.7 | 52.9 | 81.6 | 84.7 | 53.0 | 36.1 | 48.1 | 76.6 |
| 0.1 | 72.4 | 94.9 | 72.4 | 99.3 | 91.3 | 88.2 | 54.3 | 88.4 | 96.3 | 87.1 | 75.6 | 81.6 | 64.3 | 53.0 | 81.7 | 86.9 | 53.6 | 35.9 | 44.5 | 77.1 |
| 0.3 | 74.9 | 94.8 | 73.0 | 99.4 | 91.3 | 88.4 | 56.3 | 88.3 | 96.2 | 87.7 | 76.1 | 77.7 | 62.8 | 54.4 | 81.0 | 87.4 | 54.8 | 35.4 | 43.9 | 77.3 |
| 0.5 | 76.5 | 94.6 | 74.1 | 99.3 | 91.9 | 88.5 | 57.4 | 88.1 | 96.6 | 88.4 | 76.1 | 74.8 | 62.5 | 54.5 | 82.8 | 87.8 | 55.6 | 35.2 | 43.5 | 77.5 |
| 0.7 | 77.6 | 93.8 | 74.2 | 99.3 | 92.5 | 88.7 | 58.3 | 88.6 | 96.4 | 88.3 | 76.3 | 71.3 | 62.3 | 54.5 | 80.7 | 87.3 | 56.1 | 33.3 | 41.8 | 77.3 |
| 0.9 | 76.2 | 91.5 | 73.2 | 99.3 | 92.2 | 85.0 | 58.1 | 86.6 | 96.1 | 87.4 | 76.8 | 61.6 | 57.5 | 51.5 | 79.6 | 65.4 | 54.2 | 26.8 | 36.7 | 74.4 |

## A.7 ATTENTION MAPS ON ANOTHER MODEL

Here, we provide additional attention-map visualizations of another model fine-tuned on Caltech 101. In Fig.7, we can reach the same conclusion as mentioned in the **Feature Attention-map Visualization** section, which demonstrates the generalizability of our method's effectiveness across different datasets.

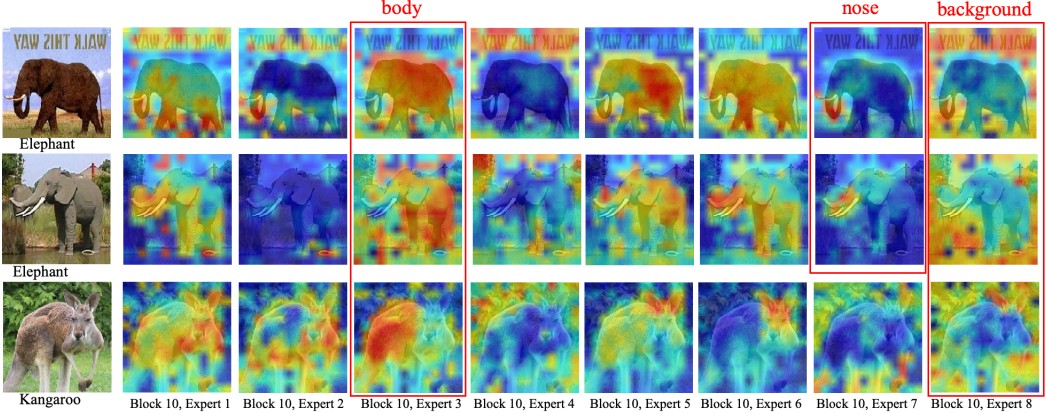

Figure 7: **Visualization of feature maps within Block 10 of Caltech101 model, Experts 1-8, across different samples.** The top two rows show the attention patterns of the same experts for two different images of elephants, highlighting consistent specialization in focusing on the body, nose, and background across similar samples. The bottom row shows the feature maps for a kangaroo image, illustrating that the attention patterns of the same experts may vary when applied to a different class of samples. These results indicate that while the same experts' attention patterns can differ substantially for different classes, they consistently focus on similar features within the same class.

