# OpenReview forum: "MLAE: Masked LoRA Experts for Visual Parameter-Efficient Fine-Tuning"
_ICLR.cc/2025/Conference — Submitted to ICLR 2025_

### Official Review · Reviewer_mLKd · 2024-10-27

**Soundness:** 3
**Presentation:** 3
**Contribution:** 2
**Rating:** 5
**Confidence:** 4

**Summary:**

The authors propose a novel approach to LoRA adapters by treating each rank as an independent expert, incorporating a rank-wise dropout mechanism to enhance diversity and independence among these experts. They also introduce a learnable weight for each rank, allowing adaptive emphasis on different ranks during training. This method, validated on VTAB-1K and FGVC datasets, demonstrates its effectiveness by achieving state-of-the-art performance with greater parameter efficiency.

**Strengths:**

1.The paper introduces a dropout mechanism specifically for LoRA adapters, applying rank-wise dropout to improve model generalizability.
2.The concept of treating each rank in the LoRA adapter as an independent expert is innovative, providing a fresh perspective on parameter-efficient fine-tuning.
3.The paper is well-structured and easy to follow, making the proposed methodology accessible and clear.

**Weaknesses:**

1.The claims regarding the use of cellular decomposition to impose independence and diversity constraints (lines 18-21, 161) lack clarity. Specifically, there is no explicit mention of how independence among the 8$\times$rank-1 LoRA matrices is achieved or how it is better than a rank-8 LoRA. Additionally, it’s not evident how using 8$\times$rank-1 LoRAs differs from a single Rank-8 LoRA, as rank 8 LoRA can be viewed as concatenate row and column vectors of the 8 rank-1 LoRAs. This raises questions about whether the observed differences in Table 6 results are significant or coincidental, and further analysis or confidence intervals would strengthen this claim.

2.Apart from the rank-wise dropout, the advantage of adding adaptive weights to each rank in the LoRA adapter appears limited, suggesting that its contribution to performance may be minor.

3.The paper does not provide sufficient configuration details for the VTAB-1K experiment, making reproducibility challenging.

4.The core idea shares similarities with recent papers, including "LoRA Dropout as a Sparsity Regularizer for Overfitting Control" (Lin et al., 2024) and "LoRA Meets Dropout under a Unified Framework" (Wang et al., 2024). Highlighting the distinctions more clearly would strengthen the paper's originality.

5.(minor)In Table 1, the source for certain numbers should be specified, particularly those taken from previous publications, like the GLoRA paper.

**Questions:**

Need either analysis on the claim for preferring 8$\times$Rank-1 LoRA against Rank-8 LoRA without the proposed dropout, or the p-value of Tab.6.
Also please provide the experiment settings, especially the configuration of the proposed method, for the VTAB-1k experiment, including which hyperparameter you choose and also the masking schedule you use to reproduce the results.

---

> ### Author Response · Authors · 2024-11-20
> **Official Response to Reviewer mLKd**
>
> We thank the reviewer mLKd for the careful reviews and constructive suggestions. We answer the questions as follows.
>
> ------
>
> > **W#1 & Q#1.1:** The claims regarding the use of cellular decomposition to impose independence and diversity constraints lack clarity. Specifically, there is no explicit mention of how independence among the 8×rank-1 LoRA matrices is achieved or how it is better than a rank-8 LoRA. Additionally, it’s not evident how using 8×rank-1 LoRAs differs from a single Rank-8 LoRA, as rank 8 LoRA can be viewed as concatenate row and column vectors of the 8 rank-1 LoRAs. This raises questions about whether the observed differences in Table 6 results are significant or coincidental, and further analysis or confidence intervals would strengthen this claim.
>
> **A#1:** We sincerely apologize for the confusion. As you correctly pointed out, a rank-8 LoRA can indeed be viewed as the concatenation of row and column vectors from 8 rank-1 LoRAs, and they are theoretically equivalent. What we want to clarify is that cellular decomposition does not directly impose independence and diversity. The key mechanism at work is the random masking implemented through expert-level dropout. In fact, cellular decomposition serves as the foundation for applying masking. Without cellular decomposition, directly masking the parameters of the entire weight matrix would neither reduce parameter similarity nor improve performance.
>
> All the results in Table 6 use the same hyperparameter configuration as MLAE. Detailed configurations can be found in the appendix **A.2 IMPLEMENTATION DETAILS** of the paper, specifically in **Table 10.** While 8_1, 4_2, and 2_4 remain theoretically equivalent after cellular decomposition, they become non-equivalent when combined with expert-level dropout. Therefore, the observed differences in results are reasonable.
>
> ------
>
> > **W#2:** Apart from the rank-wise dropout, the advantage of adding adaptive weights to each rank in the LoRA adapter appears limited, suggesting that its contribution to performance may be minor.
>
> **A#2:** Thank you for your constructive comments. While introducing adaptive coefficients only improves the average performance by 0.2% across 19 datasets, we observed significant performance gains on several individual datasets. Moreover, the parameter size of the adaptive coefficients is extremely small, resulting in negligible additional memory and computational overhead during training. Therefore, we believe that adopting adaptive coefficients is a worthwhile enhancement.
>
> ------
>
> > **W#4:** The core idea shares similarities with recent papers, including "LoRA Dropout as a Sparsity Regularizer for Overfitting Control" and "LoRA Meets Dropout under a Unified Framework". Highlighting the distinctions more clearly would strengthen the paper's originality.
>
> **A#3:** Thank you for your detailed and constructive comments. In the original work "LoRA Dropout as a Sparsity Regularizer for Overfitting Control", it is mentioned that "**we conduct dropout on the input/output dimension of both matrices.**" In contrast, our expert-level dropout applies dropout on the **rank dimension**. We also want to emphasize that while we use dropout to implement masking, our key contribution lies in **introducing masking based on cellular decomposition**. Expert-level dropout is merely one method for achieving random masking. And the primary contribution of "LoRA Meets Dropout under a Unified Framework" is the introduction of the **HiddenKey** method, which modifies transformer architectures via dropout to address what they claim to be LoRA’s susceptibility to overfitting. However, it is important to note that their work does not involve structural improvements to the LoRA framework itself.
>
> ------
>
> > **W#5:** (minor)In Table 1, the source for certain numbers should be specified, particularly those taken from previous publications, like the GLoRA paper.
>
> **A#4:** Thank you for your thoughtful suggestions. We will explicitly specify the sources of all experimental results in the revised version.
>
> ------
>
> > **W#3 & Q#1.2:**  Also please provide the experiment settings, especially the configuration of the proposed method, for the VTAB-1k experiment, including which hyperparameter you choose and also the masking schedule you use to reproduce the results.
>
> **A#5:** Thank you for your insightful comments. We ultimately adopted stochastic masking, and all experts within each block of the model use the same drop rate. For different datasets, we employed varying initialization coefficients and drop rates. For more comprehensive configurations, please refer to the appendix of the paper: **A.1 Dataset Details**, **A.2 Implementation Details**, and **A.5 Configurations of Masking Strategies**. If you have any further questions about the experimental settings, please do not hesitate to let us know.

---

> ### Author Response · Authors · 2024-11-24
> **Follow up**
>
> > Q#1: Have you tried to perform ablation on the Mask and Mask+Adaptive on the Ablation studies for MLAE? i.e. applying dropout only after the LoRA output with Mask or Mask+Adaptive?
>
> **A#1:** Thank you for your prompt response. First, we would like to clarify the implementation of LoRA+dropout mentioned in our paper. **Essentially, LoRA+dropout corresponds to the ablation experiment you suggested in your previous feedback, which involves applying dropout only after the LoRA output.** The detailed pseudocode is as follows:
>
> ```python
> #LoRA
> result = (self.lora_dropout(x) @ self.lora_A.T @ self.lora_B.T) * self.scaling
> #LoRA+dropout
> result = (self.lora_dropout(x) @ F.dropout(self.lora_A.T @ self.lora_B.T, p=self.drop_rate, training=self.training)) * self.scaling
> ```
>
> ------
>
> We would also like to provide an explanation of the LoRA dropout method proposed in *"LoRA Dropout as a Sparsity Regularizer for Overfitting Control"*. The specific formula is as follows:
> $$
> \hat{A} = A \cdot \text{diag}(m_A), \quad m_A \sim \text{Bern}(1 - p); \\
> \hat{B} = (B \cdot \text{diag}(m_B))^T, \quad m_B \sim \text{Bern}(1 - p);\\
> \hat{h} = W_0 x + \hat{B} \hat{A} x.
> $$
> In this method, dropout is applied to both the input and output dimensions using the `diag(.)` operation, rather than the rank dimension. **This approach differs from the LoRA+dropout method referenced in our work.** We apologize for any confusion caused by the overlap in terminology, as these are concurrent works. We will address this distinction more explicitly in the revised version of our work.
>
> ------
>
> > Q#2: Also, for Figure 4, I would like to ask the authors why LoRA+Dropout is introduced in the calculation of parameter cosine similarity other than accuracy? If possible, please include the LoRA+Dropout accuracy metrics.
>
> **A#2:** We sincerely appreciate your suggestion. In response, we have conducted additional experiments to compare MLAE and LoRA+dropout (using the same initialization coefficients as MLAE) on the VTAB-1k benchmark. The specific results are summarized in the table below. We will also include these results in the revised version of our paper.
>
> |                | Natural | Specialized | Structured | Mean Acc. |
> | -------------- | :-----: | :---------: | :--------: | :-------: |
> | LoRA + dropout |  81.9   |    86.2     |    62.9    |   77.0    |
> | MLAE           |  **84.0**   |    **87.5**     |    **64.8**    |   **78.8**    |

---

> ### Author Response · Authors · 2024-11-25
> **Supplementary Experimental Results**
>
> > Q#3: Also, can you compare your method with the above mentioned papers (i.e. LoRA Dropout and LoRA meets Dropout)?
>
> **A#3:**  We sincerely appreciate your constructive feedback. In the table below, we compare the experimental results of *LoRA Dropout* and MLAE on the VTAB-1k benchmark (*LoRA Dropout* results are sourced from their original paper). Since *LoRA meets Dropout* primarily focuses on fine-tuning large language models (LLMs) and did not conduct experiments on the VTAB-1k benchmark, and as the code for this work has not been publicly released, it is challenging for us to directly reproduce their results.
>
> |                | Natural  | Specialized | Structured | Mean Acc. |
> | -------------- | :------: | :---------: | :--------: | :-------: |
> | *LoRA Dropout* |   81.7   |  **90.2**   |    63.8    |   78.6    |
> | MLAE           | **84.0** |    87.5     |  **64.8**  | **78.8**  |
>
> From the table, it is evident that MLAE consistently outperforms *LoRA Dropout* on datasets of both the **Natural** and **Structure** types, and also achieves a higher average accuracy. Notably, *LoRA Dropout* samples multiple dropout instances during both training and testing, which **introduces considerable computational overhead**. In contrast, MLAE does not employ any such sampling operations. As a result, MLAE achieves better performance under the same training cost as *LoRA* (refer to **Official Response to Reviewer SyV9: A#2**) while being simpler and more efficient.

---

### Official Review · Reviewer_SyV9 · 2024-10-29

**Soundness:** 3
**Presentation:** 3
**Contribution:** 2
**Rating:** 5
**Confidence:** 4

**Summary:**

The paper addresses the redundancy and limited diversity in LoRA's low-rank matrices. The authors introduce Masked LoRA Experts (MLAE), which decomposes low-rank matrices into rank-1 "experts" and applies masking techniques to selectively activate these experts during training. The method involves Cellular Decomposition to divide low-rank matrices into independent rank-1 submatrices or experts, followed by masking. Meanwhile, adaptive coefficients for each expert are determined to control their contribution based on their importance. Experiments conducted on the VTAB-1k and FGVC benchmarks demonstrate that MLAE surpasses previous parameter-efficient fine-tuning baselines.

**Strengths:**

+ While the LoRA framework has been used within the mixture of experts (MoE) before, the use of masking at the expert level sounds novel. The authors demonstrate how the adaptive dropout technique combined with cellular decomposition effectively addresses redundancy.
+ The authors provide extensive experiments across 24 tasks covering both general vision tasks and fine-grained classification.
+ The paper is well-organized and easy to follow.

**Weaknesses:**

- Regarding Masking Configurations: While stochastic masking performed best, the analysis could be expanded by exploring additional dropout probabilities or scheduling strategies to better understand how different levels of sparsity affect each dataset type (e.g., specialized vs. structured).

- Computational Costs: The authors briefly mention longer training times for MLAE due to the introduction of dropout but do not fully explore how significant this increase is. Adding a thorough analysis of training time vs. performance trade-offs could improve the practical usability of MLAE.

**Questions:**

- In the experiments, stochastic masking is found to perform best. Have different dropout probabilities or dynamic dropout scheduling been considered within each masking type?
- Could the authors provide more insight into why cellular decomposition (rank-1 expert decomposition) specifically benefits generalization? Is there a theoretical foundation for why smaller, rank-1 updates lead to better performance in PEFT, or is it primarily an empirical observation?

---

> ### Author Response · Authors · 2024-11-20
> **Official Response to Reviewer SyV9**
>
> We thank the reviewer SyV9 for the careful reviews and constructive suggestions. We answer the questions as follows.
>
> ------
>
> > **W#1**: Regarding Masking Configurations: While stochastic masking performed best, the analysis could be expanded by exploring additional dropout probabilities or scheduling strategies to better understand how different levels of sparsity affect each dataset type (e.g., specialized vs. structured).
>
> **A#1:** Thank you very much for your constructive suggestion. In the table below, we present the results of different scheduling strategies under **stochastic masking** for each dataset type, as well as the results for different drop rates $p$ (without the best initialization) for each dataset type. For more detailed data, please refer to **Table 14** and **Table 15** in the appendix of the paper. From the results, we observe the following:
>
> 1. Regarding the Scheduling Strategy, the MLAE method achieves the best performance across all three dataset types.
> 2. Regarding Drop Rate $p$, **Natural** and **Specialized** dataset types tend to prefer higher drop rates, while the **Structured** dataset type tends to perform better with lower drop rates.
>
> | Scheduling Strategy | Natural  | Specialized | Structured | Average  |
> | :------------------ | :------: | :---------: | :--------: | :------: |
> | Incremental         |   82.9   |    87.2     |    63.0    |   77.7   |
> | Hourglass           |   82.7   |    86.3     |    59.8    |   76.3   |
> | Decremental         |   82.5   |    85.7     |    59.6    |   76.0   |
> | Protruding          |   81.5   |    87.3     |    64.1    |   77.6   |
> | MLAE                | **84.0** |  **87.5**   |  **64.8**  | **78.8** |
>
> | $p$  | Natural  | Specialized | Structured | Average |
> | :--: | :------: | :---------: | :--------: | :-----: |
> | 0.0  |   80.9   |    86.0     |  **62.9**  |  76.6   |
> | 0.1  |   81.8   |    86.9     |    62.7    |  77.1   |
> | 0.3  |   82.6   |    87.1     |    62.2    |  77.3   |
> | 0.5  |   83.2   |    87.3     |    62.1    |  77.5   |
> | 0.7  | **83.5** |  **87.4**   |    60.9    |  77.3   |
> | 0.9  |   82.2   |    87.2     |    54.2    |  74.4   |
>
> ------
>
> > **W#2**: Computational Costs: The authors briefly mention longer training times for MLAE due to the introduction of dropout but do not fully explore how significant this increase is. Adding a thorough analysis of training time vs. performance trade-offs could improve the practical usability of MLAE.
>
> **A#2:** Thank you for your detailed and constructive comments. Encouragingly, we have recently adopted a more efficient implementation that significantly reduces training time. Using the same NVIDIA RTX A6000 under identical device conditions, we trained both the LoRA and MLAE methods on the VTAB-1k benchmark for 100 epochs per dataset. We recorded the training time for each dataset and repeated the process 5 times to calculate the average training time. Remarkably, the speed of MLAE is on par with LoRA and even demonstrates slight acceleration in certain cases. The detailed results are as follows.
>
> |      | Average | Natural | Specialized | Structured |
> | ---- | :-----: | :-----: | :---------: | :--------: |
> | LoRA |  1149s  |  912s   |    1179s    |   1343s    |
> | MLAE |  1133s  |  905s   |    1177s    |   1312s    |
>
> ------
>
> > **Q#1:** In the experiments, stochastic masking is found to perform best. Have different dropout probabilities or dynamic dropout scheduling been considered within each masking type?
>
> **A#3:** Thank you for your detailed comments. We have taken your suggestions into account when implementing different masking types. For example, in the incremental pattern, the dropout probability assignment from block 1 to block 12 is set as [0.0, 0.1, 0.2, 0.3, 0.4, 0.5, 0.6, 0.6, 0.7, 0.7, 0.8, 0.8]. For detailed configurations of other masking types, please refer to **Appendix A.5: Configurations of Masking Strategies** in the paper.
>
> ------
>
> > **Q#2:** Could the authors provide more insight into why cellular decomposition (rank-1 expert decomposition) specifically benefits generalization? Is there a theoretical foundation for why smaller, rank-1 updates lead to better performance in PEFT, or is it primarily an empirical observation?
>
> **A#4:** Thank you for your detailed comments. We would like to clarify that cellular decomposition **does not directly benefit generalization**; the key mechanism at work is the random masking implemented through expert-level dropout. In fact, cellular decomposition serves as the foundation for masking. Without cellular decomposition, directly applying masking to the parameters of the weight matrix would neither reduce parameter similarity nor improve performance. Regarding why MLAE leads to better performance, we have provided the theoretical intuition in **Section 4.3, lines 375–399** of the paper.

---

> > ### Comment · Reviewer_SyV9 · 2024-11-24
> >
> > I've read the authors' responses and other reviewers' comments. The authors have addressed most of my main concerns, but I’d like to see other reviewers’ feedback on their points before making a final decision.

---

> ### Author Response · Authors · 2024-12-02
>
> We sincerely appreciate your feedback and your acknowledgment of our efforts to address the previous comments. However, we would like to clarify that, contrary to your statement that “the proposed framework is quite complex,” we believe our proposed framework is actually quite simple and efficient, especially when compared to similar works such as AdaLoRA [1] and IncreLoRA [2].
>
> To further alleviate your concerns, we are more than willing to provide anonymous access to our code, which we hope will demonstrate the simplicity and effectiveness of our approach.
>
> We kindly request that you reconsider evaluating our work with this additional information.
>
> [1] Zhang et al. AdaLoRA: Adaptive Budget Allocation for Parameter-Efficient Fine-Tuning. ICLR 2023
>
> [2] Zhang et al. IncreLoRA: Incremental Parameter Allocation Method for Parameter-Efficient Fine-tuning

---

> > ### Comment · Reviewer_SyV9 · 2024-12-02
> >
> > Thank you for your response. Yes, please provide anonymous access to your code to help assess the simplicity and effectiveness of your approach further.

---

> > > ### Author Response · Authors · 2024-12-02
> > >
> > > We greatly appreciate the time and effort you’ve dedicated to reviewing our code. The provided anonymous code link is updatable, and if you need additional code to aid in your understanding, we would be happy to update the link.
> > >
> > > Additionally, for the reproducibility of our work, we will release the complete code upon acceptance.

---

> ### Author Response · Authors · 2024-12-02
>
> Thank you very much for your timely response. We have provided the main code at the following link: https://anonymous.4open.science/r/MLAE-C83C.
>
> If you have any further questions or concerns regarding the code, please feel free to reach out to us.

---

### Official Review · Reviewer_w29t · 2024-11-04

**Soundness:** 4
**Presentation:** 4
**Contribution:** 3
**Rating:** 6
**Confidence:** 5

**Summary:**

The paper discusses an improvement to Low-Rank Adaptation (LoRA), called Masked LoRA Experts (MLAE). The key idea is to randomly drop submatrices within the low-rank matrix during fine-tuning. The authors claim that such a strategy can enhance learning diversity and parameter quality. Moreover, they conduct multiple experiments on visual recognition, including VTAB-1k and FGVC, to demonstrate the effectiveness of MLAE.

**Strengths:**

1.	This paper is clearly presented and well-organized. The authors also provide a detailed discussion of related works and variants.
2.	MLAE compares multiple masking strategies, including fixed, random, and mixed masking, and presents detailed experimental results for reference.
3.	MLAE shows commendable performance across multiple datasets, surpassing GLoRA on most datasets involved.

**Weaknesses:**

1.	Two main components, the mix of LoRA experts and MoE dropout, have been discussed in previous studies [1,2], thereby limiting the novelty and technical contribution of MLAE.
2.	Since MLAE shares some similarities with recent related competitive baselines, both the differences and performance should be compared between MLAE and these works (e.g., IncreLoRA)
3.	Given that this article proposes a task-independent LoRA improvement, it would be better to perform comparisons with existing LoRA-based methods on fine-tuning LLMs, the main benchmark of PEFT methods.
[1]. Gao et al. Higher layers need more lora experts. arXiv preprint arXiv:2402.08562, 2024.
[2] Chen et al. Sparse moe as the new dropout: Scaling dense and self-slimmable transformers. arXiv preprint arXiv:2303.01610, 2023.

**Questions:**

1.	In Section 3.3 Adaptive Coefficients, I know that \lambda is set as a learnable parameter, but the paper does not explicitly state this. Suggest clarification.
2.	The ablation experiment in Table 5 seems to indicate that the effect of adaptive coefficients is minimal, e.g., only a 0.2% improvement on average. Why introduce this?
3.	Table 10 indicates that each dataset requires searching for appropriate hyperparameters. How sensitive is the model to different parameters?
4.	Why is lora+dropout not more effective than MLAE?

---

> ### Author Response · Authors · 2024-11-21
> **Official Response to Reviewer w29t 1/3**
>
> We thank the reviewer w29t for the careful reviews and constructive suggestions. We answer the questions as follows.
>
> ------
>
> > **W#1:** Two main components, the mix of LoRA experts and MoE dropout, have been discussed in previous studies [1,2], thereby limiting the novelty and technical contribution of MLAE.
>
> **A#1:** Thank you for your insightful comments. We would like to clarify that our primary contribution lies in **introducing the concept of masking to the fine-tuning domain**. As **Reviewer SyV9** pointed out, **"While the** **LoRA** **framework has been used within the mixture of experts (MoE) before, the use of masking at the expert level is novel."**  Our results show that random masking performs the best, with expert-level dropout being just one of its implementations. We believe that other implementation methods or alternative masking approaches, such as those based on parameter importance scores, have the potential to achieve even better results. We are committed to further investigation in this direction—please stay tuned!
>
> ------
>
> > **W#2:** Since MLAE shares some similarities with recent related competitive baselines, both the differences and performance should be compared between MLAE and these works (e.g., IncreLoRA)
>
> **A#2:**  Thank you for your constructive suggestions. We would like to clarify that, under a given parameter budget, while the ultimate goal of MLAE, AdaLoRA, and IncreLoRA is to obtain high-quality parameters, the approaches differ significantly. AdaLoRA and IncreLoRA dynamically adjust the ranks of low-rank matrices during training by computing parameter importance scores, effectively discarding or adding rank-1 LoRA experts permanently. In contrast, MLAE introduces a masking mechanism to randomly deactivate certain experts during training, reducing parameter similarity and thus enhancing parameter quality. Compared to AdaLoRA and IncreLoRA, our method avoids the potential instability caused by permanently discarding experts and reduces the computational overhead associated with calculating parameter importance scores. To further illustrate the advantages of our approach, we have added a comparative experimental analysis on the VTAB-1k benchmark between MLAE and the more widely adopted AdaLoRA in the table below.
>
> |         | Natural | Specialized | Structured | Mean Acc. |
> | ------- | :-----: | :---------: | :--------: | :-------: |
> | AdaLoRA |  83.0   |    86.5     |    62.0    |   77.2    |
> | MLAE    |  **84.0**   |    **87.5**     |    **64.8**    |   **78.8**    |

---

> ### Author Response · Authors · 2024-11-21
> **Official Response to Reviewer w29t 2/3**
>
> > **W#3:** Given that this article proposes a task-independent LoRA improvement, it would be better to perform comparisons with existing LoRA-based methods on fine-tuning LLMs, the main benchmark of PEFT methods.
>
> **A#3:** Thank you for your detailed and insightful comments. Similar to works in the Visual PEFT domain, such as Repadapter [1], SPT [2], NoAH [3], MLAE primarily focuses on efficient fine-tuning of vision pre-trained models. Therefore, our paper mainly reports fine-tuning results on vision pre-trained models. However, to demonstrate the generalizability of MLAE, we have included additional experiments, with the results presented in the table below. Following LoRA, we fine-tuned the RoBERTa-base and RoBERTa-large models, reporting the validation results on the **GLUE benchmark**. From the table, it is evident that our method,  achieved an average score of **87.7** when fine-tuned on RoBERTa-base, which is 0.5 percentage points higher than LoRA. On RoBERTa-large, the average score reached **89.5**, outperforming LoRA.
>
> | **Model & Method**  | **#** **Trainable**  **Parameters** | **MNLI** | **SST-2** | **MRPC** | **CoLA** | **QNLI** | **QQP**  | **RTE**  | **STS-B** | **Avg.** |
> | ------------------- | :---------------------------------: | -------- | --------- | -------- | -------- | -------- | -------- | -------- | --------- | -------- |
> | RoBbase (FT)        |               125.0M                | **87.6** | 94.8      | 90.2     | 63.6     | 92.8     | **91.9** | 78.7     | 91.2      | 86.4     |
> | RoBbase (BitFit)    |                0.1M                 | 84.7     | 93.7      | **92.7** | 62.0     | 91.8     | 84.0     | 81.5     | 90.8      | 85.2     |
> | RoBbase (AdptD)     |                0.3M                 | 87.1     | 94.2      | 88.5     | 60.8     | 93.1     | 90.2     | 71.5     | 89.7      | 84.4     |
> | RoBbase (LoRA)      |                0.3M                 | 87.5     | **95.1**  | 89.7     | 63.4     | **93.3** | 90.8     | 86.6     | 91.5      | 87.2     |
> | **RoBbase (MLAE)**  |               0.305M                | **87.6** | 94.8      | 91.7     | **65.1** | 93.2     | 90.3     | **87.0** | **91.6**  | **87.7** |
> |                     |                                     |          |           |          |          |          |          |          |           |          |
> | RoBlarge (FT)       |               355.0M                | 90.2     | 96.4      | 90.9     | 68.0     | 94.7     | **92.2** | 86.6     | 92.4      | 88.9     |
> | RoBlarge (LoRA)     |                0.8M                 | 90.6     | 96.2      | 90.9     | **68.2** | 94.9     | 91.6     | 87.4     | **92.6**  | 89.0     |
> | **RoBlarge (MLAE)** |                0.81M                | **90.9** | **96.6**  | **91.2** | 68.1     | **95.2** | 91.4     | **89.9** | 92.5      | **89.5** |
>
> [1] Luo et al. Towards Efficient Visual Adaption via Structural Re-parameterization.
>
> [2] He et al. Sensitivity-Aware Visual Parameter-Efficient Fine-Tuning. ICCV 2023 oral.
>
> [3] Zhang et al. Neural Prompt Search. TPAMI.

---

> ### Author Response · Authors · 2024-11-21
> **Official Response to Reviewer w29t 3/3**
>
> > **Q#1:** In Section 3.3 Adaptive Coefficients, I know that \lambda is set as a learnable parameter, but the paper does not explicitly state this. Suggest clarification.
>
> **A#4:** Thank you very much for pointing out this issue. Indeed, $\lambda$ is a learnable parameter, and we will explicitly clarify this in the revised version.
>
> ------
>
> > **Q#2:** The ablation experiment in Table 5 seems to indicate that the effect of adaptive coefficients is minimal, e.g., only a 0.2% improvement on average. Why introduce this?
>
> **A#5:** Thank you for your constructive comments. While introducing adaptive coefficients only improves the average performance by 0.2% across 19 datasets, we observed significant performance gains on several individual datasets. Moreover, the parameter size of the adaptive coefficients is extremely small, resulting in negligible additional memory and computational overhead during training. Therefore, we believe that adopting adaptive coefficients is a worthwhile enhancement.
>
> ------
>
> > **Q#3:** Table 10 indicates that each dataset requires searching for appropriate hyperparameters. How sensitive is the model to different parameters?
>
> **A#6:** Thank you for your profound comments. We have provided the model performance corresponding to different hyperparameters in the tables below. Table 1 presents the results of different initializations on the VTAB-1K benchmark, while Table 2 shows the results of different probabilities on the VTAB-1K benchmark (**without the best initialization**). As observed from the tables, our model demonstrates good stability and robustness with respect to these two hyperparameters. For more detailed data, please refer to Table 14 and Table 15 in the appendix of the paper.
>
> | Initialization | 0.125 | 0.25 | 0.5  | 1    | 2    | 4    |
> | -------------- | ----- | ---- | ---- | ---- | ---- | ---- |
> | Acc(Average)   | 78.0  | 78.2 | 78.2 | 78.3 | 78.1 | 77.9 |
>
> | $p$          | 0.0  | 0.1  | 0.3  | 0.5  | 0.7  |
> | :----------- | ---- | ---- | ---- | ---- | ---- |
> | Acc(Average) | 76.6 | 77.1 | 77.3 | 77.5 | 77.4 |
>
> ------
>
> > **Q#4:** Why is lora+dropout not more effective than MLAE?
>
> **A#7:** Thank you for your insightful comments. We believe this might be due to the fact that individual parameter values in the weight matrix do not inherently carry semantic information. As a result, applying dropout directly to the entire weight matrix is unlikely to be meaningful. From an experimental perspective, as shown in Figure 4 of the paper, LoRA+dropout does not significantly reduce the similarity between experts, and therefore it fails to make the experts more independent or to encourage learning more diverse information.

---

> ### Author Response · Authors · 2024-11-29
> **Thanks for Reviewer w29t's valuable feedback.**
>
> Dear Reviewer w29t,
>
> **Thank you for your positive feedback and the increased score.**
>
> We are delighted that our response has successfully addressed your concerns. Your suggestions have been incredibly insightful and have greatly enhanced the quality of our work. We sincerely thank you once again for taking the time to review our rebuttal and for your valuable feedback.
>
> Thanks.
>
> Authors

---

### Author Response · Authors · 2024-11-21
**General Response**

We thank the reviewers for their time and thoughtful feedback. We appreciate the positive comments from the reviewers, such as w29t's observation that our paper **"shows commendable performance across multiple datasets,"** SyV9's acknowledgment that **"the use of masking at the expert level sounds novel,"** and mLKd's remark that **"the concept of treating each rank in the LoRA adapter as an independent expert is innovative, providing a fresh perspective on parameter-efficient fine-tuning."**

We have carefully addressed each identified weakness and question from all reviewers under their respective reviews. Please feel free to let us know if you have any lingering questions or if there are additional clarifications we can provide during the discussion period to further improve your rating of our paper.

---

### Author Response · Authors · 2024-12-03
**Summary of Revision and Discussion**

Dear Program Chairs (PC), Senior Area Chairs (SAC), Area Chairs (AC), and Reviewers,

As the discussion phase has ended, we would like to extend our heartfelt gratitude to everyone who participated in the discussion. Your insights and questions have been invaluable, and we have made significant efforts to address all the issues you raised.

We have provided additional analyses **(Different Masking Configurations, Thorough Analysis of Computational Costs)**, introduced new baselines **(GLUE benchmark, AdaLoRA, IncreLoRA, LoRA_dropout)**, and offered detailed explanations to clarify any potential ambiguities, alongside the main anonymized code **(https://anonymous.4open.science/r/MLAE-C83C).**

**We are delighted that our response has successfully addressed reviewer w29t's concerns, resulting in an improvement in the score.** While we regret not receiving further feedback from some reviewers after implementing these updates, we hope that our efforts demonstrate our commitment to improving the paper. **We sincerely hope that you will take our revisions and explanations into account during your final evaluations.** Your feedback has been instrumental in enhancing the quality of our work, and we deeply appreciate the time and effort you have dedicated to reviewing our submission. Once again, thank you for your invaluable contributions to our work.

Best regards,

Authors

---

### Meta-Review · Area_Chair_T9uR · 2024-12-17

**Metareview:**

The paper proposes an new approach that applies the concept of masking to visual PEFT. However, according to the reviewers’ comments, the technical methods employed in this research do not differ significantly from those of predecessors, thereby limiting its innovation. The overall contribution does not seem substantial for publication at ICLR.

**Additional Comments On Reviewer Discussion:**

The authors have provided additional analyses, introduced new baselines, and offered detailed explanations to clarify the ambiguities.  Reviewer w29t has raised the score.
Reviewer SyV9 still believes that the overall contribution does not seem substantial.
The final average score is still below the accept threshold.

---

### Decision · Program_Chairs · 2025-01-22

Reject